# JenTab: A Toolkit for Semantic Table Annotations

Nora Abdelmageed[1−4][0000−0002−1405−6860] and Sirko Schindler[2,4][0000−0002−0964−4457]

[1] Computer Vision Group
[2] Heinz Nixdorf Chair for Distributed Information Systems
[3] Michael Stifel Center Jena, Germany
[4] Friedrich Schiller University Jena, Germany
`nora.abdelmageed@uni-jena.de, sirko.schindler@uni-jena.de`

**Abstract.** Tables are a ubiquitous source of structured information. However, their use in automated pipelines is severely affected by conflicts in naming and issues like missing entries or spelling mistakes. The Semantic Web has proven itself a valuable tool in dealing with such issues, allowing the fusion of data from heterogeneous sources. Its usage requires the annotation of table elements like cells and columns with entities from existing knowledge graphs. Automating this semantic annotation, especially for noisy tabular data, remains a challenge, though. JenTab is a modular system to map table contents onto large knowledge graphs like Wikidata. It starts by creating an initial pool of candidates for possible annotations. Over multiple iterations context information is then used to eliminate candidates until, eventually, a single annotation is identified as the best match. Based on the SemTab2020 dataset, this paper presents various experiments to evaluate the performance of JenTab. This includes a detailed analysis of individual components and of the impact different approaches. Further, we evaluate JenTab against other systems and demonstrate its effectiveness in table annotation tasks.

**Keywords:** knowledge graph, matching, tabular data, semantic annotation

## 1 Introduction

Although a considerable amount of data is published in tabular form, oftentimes, the information contained is hardly accessible to automated processes. Causes range from issues like misspellings and partial omissions to the ambiguity introduced by using different naming schemes, languages, or abbreviations. The Semantic Web promises to overcome the ambiguities but requires annotation with semantic entities and relations. The process of annotating a tabular dataset to a given Knowledge Graph (KG) is called Semantic Table Annotation (STA). The objective is to map individual table elements to their counterparts from the KG as illustrated in Figure 1 (naming according to [11]): Cell Entity Annotation (CEA) matches cells to individuals, whereas Column Type Annotation (CTA) does the same for columns and classes. Furthermore, Column Property Annotation (CPA) captures the relationship between pairs of columns.

JenTab is a toolkit to annotate large corpora of tables. It follows a general pattern of Create, Filter and Select (CFS): First, for each annotation, initial candidates are generated using appropriate lookup techniques (Create). Subsequently, the available context is used in multiple iterations to narrow down these sets of candidates as much

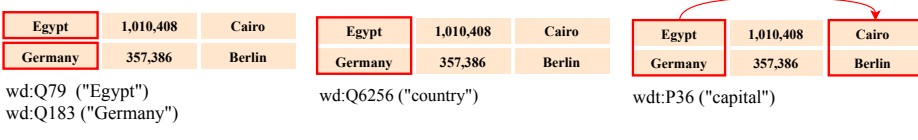

wd:Q79 ("Egypt")
wd:Q183 ("Germany")

wd:Q6256 ("country")

wdt:P36 ("capital")

(a) Cell annotation.          (b) Column annotation.          (c) Property annotation.

Fig. 1: Illustration of Semantic Table Annotation (STA) tasks[6].

as possible (Filter). Finally, if multiple candidates remain, a solution is chosen among them (Select). We provide several modules for each of these steps. Different combinations allow to fine-tune the annotation process by considering both the modules' performance characteristics and their impact on the generated solutions. The contributions of our paper are as follows. All experiments are based on the large corpus provided by Semantic Web Challenge on Tabular Data to Knowledge Graph Matching (SemTab2020) [8,10,11][5] ($\sim 130,000$ tables) matching the content to Wikidata [21].

– We demonstrate the effectiveness of JenTab relying only on publicly available lookup services.
– We provide a detailed evaluation of the impact individual modules have on the candidate generation.
– We perform three experiments exploring different CTA-strategies that vary the mode of determining cells' types and hence the column annotation.
– We compare JenTab's performance to other top contenders of the SemTab2020.

The remainder of this paper is structured as follows. Section 2 gives an overview of the related work. Section 3 describes our pipeline. Section 4 explains the dataset, encountered challenges, and the metrics used in our evaluation. Section 5 discusses our experiments and results. Section 6 concludes the paper and shows future directions.

## 2    Related Work

We start by briefly reviewing benchmark datasets and motivate the selection of the SemTab2020 dataset for our evaluation. We then summarize existing approaches to match tabular data to KGs. While both semi-automatic and full-automatic approaches have been proposed, we will focus our attention on later ones. This is in line with the assumptions in this paper and the conditions posed by the SemTab challenges.

*Benchmarks.* In the past, various benchmarks have been proposed and used for STA tasks. Manually annotated corpora like T2Dv2[7] or the ones used in [3,15] offer only a minimal number of tables. On the other hand, larger corpora are often automatically created using web tables as a source. The resulting Ground Truth (GT) data is thus rather noisy as seen, e.g., in [9]. The tables in the SemTab2020 datasets [8, 10] are artificially created from Wikidata [21]. This inverts older approaches of benchmarks creation and provides a large corpus of tables with high-quality GT data. Further, it allows adjusting the difficulty of tasks by varying the noise introduced to the tables.

---

[5] http://www.cs.ox.ac.uk/isg/challenges/sem-tab/
[6] We use the prefixes `wd:` and `wdt:` for http://www.wikidata.org/entity/ and http://www.wikidata.org/prop/direct/ respectively.
[7] http://webdatacommons.org/webtables/goldstandardV2.html

*Approaches.* ColNet [5] tackles only the CTA task. It uses a Convolutional Neural Networks (CNN) trained by classes contained within a KG. The predicted annotations are combined with the results of a traditional KG. The final annotation is selected using a score that selects the lookup solutions with high confidence and otherwise resorts to the CNN predictions. Results have shown that the lookup service outperforms the CNN prediction for a larger knowledge gap. The approach has then been extended by considering other cells in the same row in a property feature vector Property to Vector (P2Vec) as an additional signal to the neural network which yields better results [6]. Efthymiou et.al [9] have a slightly different task description. They tackle row to KG entity matching. Their approach combines a lookup model, *FactBase*, with a word embedding model trained using the KG. Two variations are proposed, each succeeding in different benchmarks. Each variant uses one model as the primary source and only resorts to the other when the first does not return any result.

All these approaches rely on lookup services for their success. However, each of them addresses only a single task from STA. Moreover, they can not cope with the frequent changes of KGs since they rely on snapshots of the KG to train their respective models.

*SemTab2019.* In the year 2019, the SemTab challenge initiated to bring together the community of automatic systems for STA tasks. A four-round-dataset was released with DBpedia [2] as a target KG. Among the participants, the following systems emerged. MTab [16], the challenge winner in 2019, relies on a joint probability distribution that is updated after more information is known. Input signals include the results of various lookup services and conditional probabilities based on the row and column context. The authors mention the computational cost from the multitude of signals as a significant drawback. CSV2KG [19], achieving second place, uses an iterative process with the following steps: (i) get an entity matching using lookup services; (ii) infer the column types and relations; (iii) refine cell mappings with the inferred column types and relations; (iv) refine subject cells using the remaining cells of the row; and (v) re-calculate the column type with all the corrected annotations. Tabularisi [20], third place in 2019, also uses lookup services. Based on the returned candidates, an adapted TF-IDF score is [8] is calculated for each candidate. A combination of this score, the Levenshtein distance between cell value and candidate label, and a distance measure between cell value and the URL tokens is used to determine the final annotation. DAGOBAH [4] assumes that entities in the same column are close in the embedding space. Candidates are first retrieved using a lookup based on regular expressions and the Levenshtein distance. Afterwards, a clustering of their vector representations using the embedding is performed to disambiguate among them. The cluster with the highest row-coverage is selected and final ambiguity are resolved via a confidence score based on the row context of the candidates.

A key success factor to those systems is the use of Wikidata and Wikipedia as additional data sources. In this paper, we focus on exploiting only the target KG data sources. Therefore, we try to maximize the benefit from a given cell value and minimize our reliance on different data sources, which leads to a more straightforward system.

*SemTab2020.* The second edition of the challenge in 2020 changed the target KG to Wikidata. MTab4Wikidata [17] builds an extensive index that includes all historic revisions. Cell annotation candidates are generated using this index and a one-edit-distance algorithm. Disambiguation is done via pairwise lookups for all pairs of entities within the same row. bbw [18] relies on two core ideas. First, *SearX*[9] as a meta-

---

[8] Term Frequency-Inverse Document Frequency.

[9] https://github.com/searx/searx

lookup enabling it to search over more than 80 engines. Second, contextual matching using two features, for example, entity and property labels. The former collects results and ranks them, while the latter picks the best matches using edit-distance. SSL [13] generates a Wikidata subgraph over a table. It leverages SPARQL queries for all tasks and does not implement any fuzzy search for entities. However, it applies a crawling process through *Google* to suggest better words and thus, overcomes the problem of spelling mistakes. LinkingPark [7] has a three-module pipeline. For entity generation, it uses the Wikidata lookup API while employing an off-the-shelf spell checker. Further, its *Property Linker* module uses a fuzzy matching technique for numeric values with a certain margin. JenTab uses a similar methodology to LinkingPark for tackling spelling mistakes but with the aid of word vectors[10]. Moreover, JenTab uses the same concept of fuzzy matching for entities and properties generation.

To our knowledge, none of the these systems provided a detailed study on various solutions for STA tasks, backward compatibility across rounds, or a time analysis.

## 3   Approach

Our system's modules can be classified into one of the following three phases, which together form a Create, Filter and Select (CFS) pattern. During the Create-phase, candidates are retrieved for each requested annotation. In the Filter-phase, the surrounding context is used to reduce the number of candidates. Eventually, in the Select-phase, the final annotations are chosen among the remaining candidates. The individual modules for the same task differ in their treatment of the textual input and the context used. This causes not only differences in the accuracy of their results but also affects their performance characteristics. In the following, we explain the necessary preprocessing steps and describe the developed modules for each phase.

### 3.1   Preprocessing

Before the actual pipeline, each table is subjected to a preprocessing phase consisting of three steps: The first step aims at normalizing the cells' content. First, we attempt to fix any encoding issues using *ftfy*[11]. Further, we remove special characters like parentheses or slashes. Finally, we use regular expressions to detect missing spaces like in "*1stGlobal Opinion Leader's Summit*". In addition to the initial values, the normalized ones are stored as a cell's "clean value". In the second step, we use regular expressions to determine the datatype of each column. While our system distinguishes more datatypes, we aggregate to those having direct equivalents in KGs, i.e. `OBJECT`, `QUANTITY`, `DATE`, and `STRING`. Cells in `OBJECT`-columns correspond to entities of the KG, while the others represent literals. In the final step, we apply type-based cleaning. In general, it attempts to extract the relevant parts of a cell value for `QUANTITY` and `DATE` columns. For example, it splits the numeric value from a possibly existing unit in `QUANTITY` cells. Similarly, redundant values like "*10/12/2020 (10 Dec 2020)*" are reduced to "*10/12/2020*".

### 3.2   Annotation modules

Tabular data offers different dimensions of context that can be used to either generate annotation candidates (Create-phase) or remove highly improbable ones (Filter-Phase).

---

[10] https://www.kaggle.com/cpmpml/spell-checker-using-word2vec
[11] https://github.com/LuminosoInsight/python-ftfy

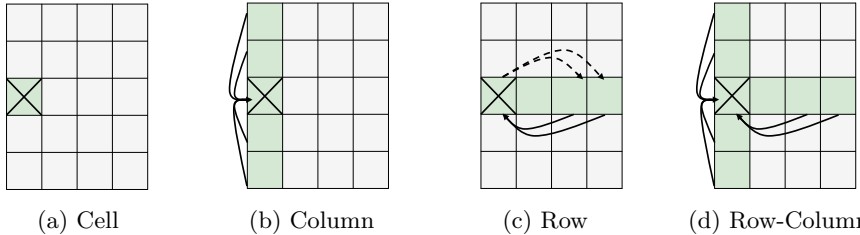

(a) Cell          (b) Column          (c) Row          (d) Row-Column

Fig. 2: Possible contexts for resolving and disambiguating annotations.

Figure 2 illustrates those visually. The *Cell Context* is the most basic one, outlined in Figure 2a. Here, nothing but an individual cell's content is available. We can then define a *Column context* as shown in Figure 2b. It is based on the premise that all cells within a column represent the same characteristic of the corresponding tuples. For the annotation process, this can be exploited insofar that all cells of a column share the same class from the KG. Annotations for cells in OBJECT-columns have further a common class as required by the CTA task. Similarly, the assumption that each row refers to one tuple leads to the *Row Context* of Figure 2c. Annotation candidates for the subject cell, i.e., a cell holding the identifier for the respective tuple/row, have to be connected to their counterparts in all other cells within the same row. Finally, all contexts can be subsumed in the *Row-Column Context* as given by Figure 2d. It combines the last two assumptions representing the most exhaustive context. In the following, we summarize our modules. For a detailed description kindly refer to [1].

*Creating Candidates* All subsequent tasks are based on suitable CEA-candidates for individual cells. The textual representation of such a cell can deviate from its canonical name and other labels given by the KG in many different ways. We devised various modules to cope with the encountered issues using the aforementioned contexts.

- **CEA Label Lookup** (*Cell Context*) employs six strategies to to cope with spelling mistakes, use of abbreviations and other lexicographical challenges.
- **CEA by column** (*Column Context*) populates the candidate pool for a cell with all available instances of that shared class.
- **CEA by subject** (*Row Context*) populates mappings for cells in the same row given the subject cell's annotation, i.e. the cell serving as an identifier for that row.
- **CEA by row** (*Row Context*) finds candidates for subject cells given the object annotations in the same row.

With candidates available for individual cells, another set of modules can be used to derive candidates for the CTA and CPA tasks.

- **CTA** collects the parent classes from all CEA-candidates for a particular column and uses them as CTA-candidates for that column.
- **CPA** retrieves all properties for CEA-candidates of subject cells and compares those to the values of the row. While object-properties are matched against the candidate lists, literal-properties use a mix of exact and fuzzy matching.
  - DATE-values are matched based on the date part omitting any additional time information. Different datetime-formats are supported.

- **STRING**-values are split into sets of tokens. Pairs with an overlap of at least 50% are considered a match.
- **QUANTITY**-values are compared using a 10% tolerance, as given in Equation 1.

$$Match = \begin{cases} true, & \text{if } |1 - \frac{value_1}{value_2}| < 0.1 \\ false, & \text{otherwise} \end{cases} \tag{1}$$

*Filtering Candidates* The previous modules generate lists of candidates for each task. Next, filter-modules remove unlikely candidates based of different contexts.

- **CTA support** (*Column Context*) removes CTA-candidates that to not apply to at least a minimum number of cells in that column.
- **CEA unmatched properties** (*Row Context*) removes CEA-candidates that are not part of any candidate for a CPA-matching.
- **CEA by property support** (*Row Context*) first counts CPA-matches for subject-cells' CEA-candidates. All but the ones scoring highest are then removed.
- **CEA by string distance** (*Cell Context*) excludes all CEA-candidates whose label is not within a certain range wrt. their Levenshtein distance [14] to the cell value.

*Selecting a Final Annotation* At some point, a final annotation from the list of candidates has to be selected. If only a single candidate is remaining, this candidate is chosen as a solution. In all other cases, the following modules will be applied.

- **CEA by string similarity** selects the CEA-candidate whose label is the closest to the original cell value using the Levenshtein distance.
- **CEA by column** operates on cells with no CEA-candidates left[12]. It looks for other cells in the same column that are reasonably close wrt. to their Levenshtein distance and adopts their solution if available.
- **CTA by LCS** considers the whole class hierarchy of current CTA-candidates and picks the Least Common Subsumer (LCS) as a solution.
- **CTA by Direct Parents** applies a majority voting on CTA-candidates and their direct parents in the class hierarchy.
- **CTA by Majority** applies a majority voting on the remaining CTA-candidates.
- **CTA by Popularity** breaks any remaining ties by selecting the most popular CTA-candidate, i.e., the one with the most instances in the KG.
- **CPA by Majority** applies a majority voting on the remaining CPA-candidates.

### 3.3   Architecture

Figure 3 shows JenTab's overall architecture. We opted for a distributed approach that allows us to split the workload across several nodes. The left-hand side depicts the two types of nodes: A central *Manager* node orchestrates a family of associated *Runner* nodes. Runners contact the Manager to request new work items, i.e., raw tables. After a work item is finished, its results are sent back to the Manager, and the next one is requested. The result of processing a single table consists of three parts:

---

[12] Under certain circumstances, the applied filter modules might have removed all CEA-candidates before.

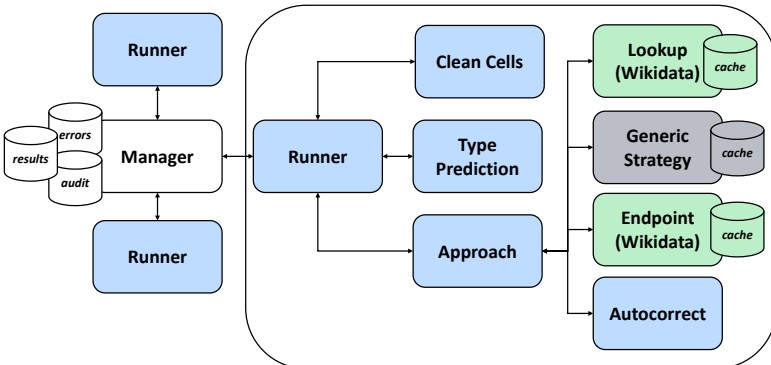

Fig. 3: JenTab system architecture.

*results* correspond to annotations of tasks, *audit data* that allows assessing the impact of individual modules, and possibly a list of any *errors* thrown during the processing.

The Manager's dashboard contains information about the following, the current state of the overall system, i.e., processed versus not yet tables, besides, data about connected Runners and errors are thrown (if any). It also gives an estimate of the remaining time needed. Finally, once the processing has finished, all gathered annotations can also be accessed from this central point. The Runner coordinates a single table's processing at a time through a series of calls to different services. Tables are first passed through the preprocessing services of *Clean Cells* and *Type Prediction*. Afterwards, the core pipeline is executed via the *Approach* service. Approach depends on the following four services. *Lookup* and *Endpoint* are proxies to the respective KG lookup and SPARQL endpoint services respectively. Moreover, the computationally expensive *Generic Strategy*, in the CEA lookup, see Subsection 3.2, is wrapped in a separate service. These three services include caching for their results. The final dependency is given by the *Autocorrect* service, which tries to fix the spellings mistakes in cells.

The chosen architecture has several advantages. First, using caches for computationally expensive tasks or external dependencies increases the overall system performance. Furthermore, it reduces the pressure on downstream systems, which is especially important when public third-party services are used. Second, when the target KG is to be substituted, all necessary changes like adjusting SPARQL queries are concentrated within just two locations: the corresponding lookup and endpoint services. Third, the distributed design allows scaling well with respect to the number of tables to be annotated. Any increase in the number of tables can be mitigated by adding new Runners to cope with the workload. Finally, the implementation allows reusable, and self encapsulated pieces of code. For example, Runner can deal with any other Approach implementation, and Autocorrect can be used by any other Approach.

## 4    Evaluation Setup

We base the evaluation of our approach on the corpus provided by the Semantic Web Challenge on Tabular Data to Knowledge Graph Matching (SemTab2020) [10]. In the following, we will first outline the configuration of annotation modules listed in Section 3, before describing the corpus in more detail. Further, we will explain the metrics used which follow the evaluation strategy prescribed by the challenge.

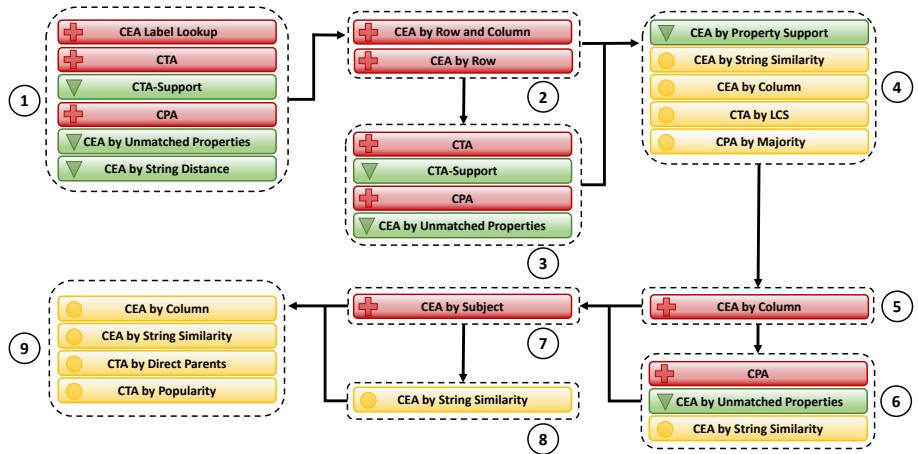

Fig. 4: JenTab: Arrangement of building blocks into a pipeline [1]. *Create* is indicated in red with a plus icon, *Filter* is represented in green and a triangle sign and *Select* is shown in yellow with a circle symbol.

## 4.1   Sequence of Modules

The order of modules used in the evaluation is outlined in Figure 4. The modules are arranged into several groups. Some groups are only executed if the preceding group had any effects on the current candidate pool. Similarly, the different approaches for creating CEA-candidates skip cells that already have candidates at the time.

Group ① represents the most direct approach. As its modules use only a few inter-dependencies, queries are rather simple and can be executed quickly. Still, it accounts for a substantial share of candidates and thus forms the basis for subsequent groups.

For cells that so far did not receive any CEA-candidates, Group ② is a first attempt to compensate by expanding the considered scope. Here, *CEA by Row and Column* precedes *CEA by Row*. Using more context information, i.e., the Column Context, returned results are of higher quality compared to *CEA by Row*. It will fail, though, when the current list of corresponding CTA-candidates do not yet contain the correct solution. In such cases, *CEA by Row* can fill in the gaps. If any of the two modules resulted in new CEA-candidates, the corresponding modules for CTA and CPA candidate creation will be repeated in Group ③.

Group ④ attempts to select annotations for the first time. A prior filter step again uses the Row Context to retain only the CEA-candidates with the highest support within their respective tuples. Afterwards, annotations are selected from the candidate pool available at this point. It yields solutions for the majority of annotation-tasks but may leave some gaps on occasion.

The next two groups represent our last efforts to generate new candidates using stronger assumptions. Group ⑤ assumes that we were already able to determine the correct CTA-annotation for the respective column and then uses all corresponding instances as CEA-candidates. Similarly, Group ⑦ assumes that the CEA-annotation subject cell is already determined and creates candidates from all connected entities.

Table 1: SemTab2020 Dataset statistics.

| Round | R1 | R2 | R3 | R4 |
|---|---|---|---|---|
| Tables # | 34,294 | 12,173 | 62,614 | 22,390 |
| Avg. Rows # ($\pm$ Std Dev.) | $7 \pm 4$ | $7 \pm 7$ | $7 \pm 5$ | $109 \pm 11,120$ |
| Avg. Columns # ($\pm$ Std Dev.) | $5 \pm 1$ | $5 \pm 1$ | $4 \pm 1$ | $4 \pm 1$ |
| Avg. Cells # ($\pm$ Std Dev.) | $36 \pm 20$ | $36 \pm 18$ | $23 \pm 18$ | $342 \pm 33,362$ |
| Target Cells # (CEA) | 985,110 | 283,446 | 768,324 | 1,662,164 |
| Target Columns # (CTA) | 34,294 | 26,726 | 97,585 | 32,461 |
| Target Columns Pairs # (CPA) | 135,774 | 43,753 | 166,633 | 56,475 |

Fig. 5: Synthetic table showcasing potential issues and challenges .

Groups ⑥, and ⑧ are used to validate those candidates and possibly select further annotations to fill in the gaps.

Group ⑨ makes a last-ditch effort for cells that could not be annotated so far. As no other module was able to find a proper solution, this group will reconsider all CEA-candidates that were dropped at some point. Using this pool, it attempts to fill the remaining gaps in annotations.

## 4.2  Dataset

We use the SemTab2020 dataset [10] as a benchmark for our approach. It contains over 130,000 tables automatically generated from Wikidata [21] that were further altered by introducing artificial noise [11]. The corpus is split into four rounds. In the last round, 180 tables are added from Tough Tables (2T) dataset [8] increasing the difficulty here. Table 1 summarizes the data characteristics of the four rounds.

Figure 5 illustrates the challenges present in the dataset. ⓐ missing or not descriptive table metadata, like column headers. ⓑ spelling mistakes. ⓒ ambiguity in cell values. For example, *UK* has (*Ukrainian (Q8798)*, *United Kingdom (Q145)*, *University of Kentucky (Q1360303)* and more) as corresponding entities in Wikidata. ⓓ missing spaces, causing tokenizers to perform poorly. ⓔ inconsistent format of date and time values. ⓕ nested pieces of information in *Quantity* fields, interfere in the corresponding CPA tasks. ⓖ redundant columns. ⓗ encoding issues. ⓘ seemingly random noise in the data. *Berlin* would be expected in the context of the given example. ⓙ missing values including nulls, empty strings or special characters like (*?*, *-*, *–*) to the same effect. ⓚ tables of excessive length.

Table 2: Generic Strategy: Unique labels and ratio of resolved labels per round.

| Rounds | Unique Labels | Matched (%) |
| --- | --- | --- |
| R1 | 252,329 | 99.0 |
| R2 | 132,948 | 98.9 |
| R3 | 361,313 | 99.0 |
| R4 | 533,015 | 96.8 |

### 4.3   Metrics

Besides the datasets, SemTab2020 also provides a framework to evaluate tabular data to knowledge graph matching systems [11]. Our evaluation follows the proposed methodology, which is outlined in the following. At its core, it relies on the standard information retrieval metrics of Precision ($P$), Recall ($R$), and F1_Score ($F1$) as given in Equation 2.

$$P = \frac{|correct\ annotations|}{|annotated\ cells|}, \ R = \frac{|correct\ annotations|}{|target\ cells|}, \ F1 = \frac{2 \times P \times R}{P + R} \quad (2)$$

However, these default metrics fall short for the CTA task. Here, there is not always a clear-cut distinction between right and wrong. Some solutions might be acceptable but do not represent the most precise class to annotate a column. Taking the last column of Figure 5 as an example, the best annotation for the last column would likely be *capital (Q5119)* (assuming "Tübingen" is noise here). Nevertheless, an annotation *city (Q515)* is also correct, but just not as precise. To account for such correct but imprecise solutions, an adapted metric called *cscore* is advised as shown in Equation 3 [12]. Here, $d(\alpha)$ is the shortest distance between the chosen annotation-entity $\alpha$ and the most precise one, i.e., the one given in the GT. Consequently, *Precision*, *Recall*, and *F1_Score* are adapted to the forms in Equation 4.

$$cscore(\alpha) = \begin{cases} 1, & \text{if } \alpha \text{ is in GT}, \\ 0.8^{d(\alpha)}, & \text{if } \alpha \text{ is an ancestor of the GT}, \\ 0.7^{d(\alpha)}, & \text{if } \alpha \text{ is a descendant of the GT}, \\ 0, & \text{otherwise} \end{cases} \quad (3)$$

$$AP = \frac{\sum cscore(\alpha)}{|annotated\ cells|}, \ AR = \frac{\sum cscore(\alpha)}{|target\ cells|}, \ AF1 = \frac{2 \times AP \times AR}{AP + AR} \quad (4)$$

## 5   Experiments and Results

In this section, we discuss our findings regarding the overall system. We start with preprocessing assessment, "Type Prediction" step which is responsible for determining a column's datatype, see Subsection 3.1. Figure 6 shows the confusion matrix of this step with 99% accuracy. We used the ground truth for CEA and CPA tasks to query Wikidata for their types; such values represent the actual datatypes, the predicted values are our system results.

Spelling mistakes are a crucial problem that we have tackled by using "Generic Strategy", see Subsection 3.2. The effectiveness of this is illustrated in Table 2: Almost 99% of unique labels were covered in the first three rounds. However, this is reduced to $\sim 97\%$ in the last round.

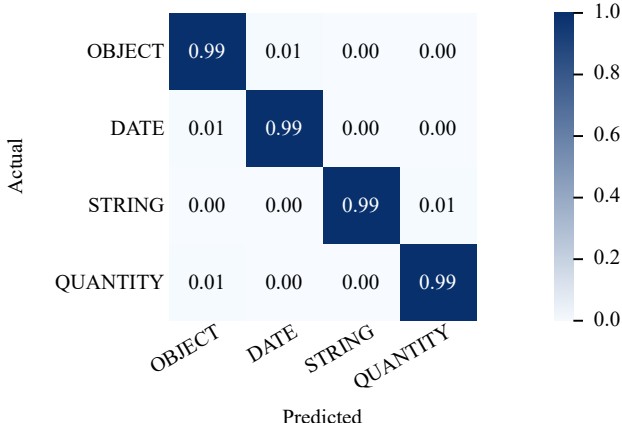

Fig. 6: Confusion matrix for type prediction. A 4-label classification task.

Our modular approach enables us to exchange individual components or provide backup solutions if the existing ones failed in specific situations. By this means, we have established three different experiments to explore the effect of changing cells' types retrieval. These three modes include: First, only direct parents through *P31*. We have used a majority vote to select a column type. Second, *2 Hops*, it includes "P31" with one parent "P279". Finally, *Multi Hops*, creates a more general tree of parents.

We have implemented five strategies for an initial CEA candidates creation, see Subsection 3.2. Figure 7a shows how much each strategy is used. This underlines the need for various strategies to capture a wide range of useful information inside each cell. The shown distribution also reflects our chosen order of methods. For example, "Generic Strategy" is our first priority, thus used most of the time. On the other hand, "Autocorrect" is has the lowest priority and is used as a means of the last resort. CEA selection phase involves two methods. Figure 7b demonstrates the use of each of them: our dominant select approach is "String Similarity", it is used by 38% more than the "Column Similarity". Finally, Figure 8a describes the distribution of CTA selection methods during the "P31" setting. While, Figure 8b represents the used methods in "2 Hops" mode, where LCS is the dominant selection strategy. Let's compare "Majority Vote" with the LCS methods in the two settings. The former successfully finds more solutions than the latter, which yields less reliance on backup strategies or tiebreakers. The same exclusive execution concept in CEA selection is also applied in CTA selection methods. The dominant method, e.g., LCS in "2 Hops" mode, is invoked more frequently due to its highest order. Other backup strategies try to solve the remaining columns if other methods failed to find a solution for them.

Table 3 reports our results for the four rounds given the three execution modes. In the first three rounds, we have reached a coverage with more than 98.8% for the three tasks. In the fourth round, CEA task, the coverage is dramatically affected by the selected mode. "P31" has achieved the highest coverage by 99.39%. However, "Multi Hops" has achieved the lowest coverage by 81.83%. $F1 - Score$ in CEA, CTA and CPA tasks is greater than 0.967, 0.945 and 0.963 receptively. We obtained these scores by

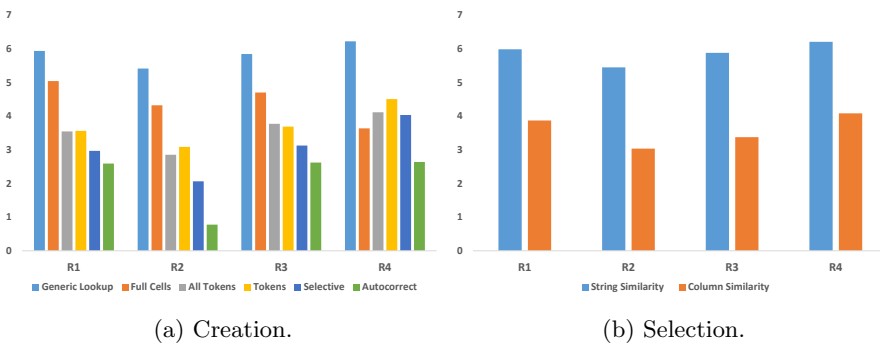

(a) Creation.                    (b) Selection.

Fig. 7: Audit statistics for CEA. y-axis is the *log* scale of the solved cells[15].

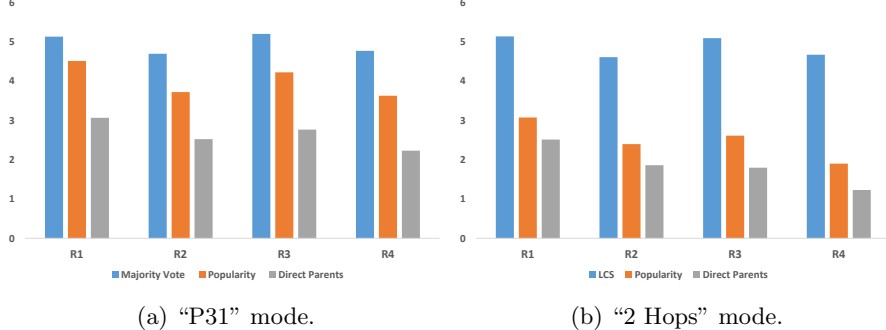

(a) "P31" mode.                 (b) "2 Hops" mode.

Fig. 8: Audit statistics for CTA selection. y-axis is the *log* scale of the solved cells.

running the publicly available evaluator code[13] on our solution files[14]. Both "2 Hops" and "Multi Hops" have better coverage but lower Recall. Unlike, "P31" which achieved the best scores in most cases. We also compare our performance with the top systems of SemTab2020 as shown in Table 4. JenTab's results are competitive across all tasks. They are severely impacted by the Tough Tables (2T) dataset [8], though.

Table 5 shows the overall system performance. It lists the time consumption for all four rounds with the number of used runners for each mode setting of the CTA task. Execution was time-scoped, i.e. an upper limit for the time per table was set. This allowed the system to converge faster compared to the initial implementation [1] with, e.g., Round 4 showing a more than 50% reduction in time. Intermediate results are cached across rounds saving time and lowering the number of requests to external services. The employed architecture allows to scale the number of runners based on available resources and hence speed up the overall process.

The results show that for most tables "P31" mode is the most efficient fastest approach. However, for the 2T dataset a more sophisticated approach is needed. Here, the "2 Hops" appraoch yields better results. The "Multiple Hops" strategy can not surpass any of the other strategies no matter the setting. In terms of both performance and results it delivers inferior results and should thus not be used.

---

[13] https://github.com/sem-tab-challenge/aicrowd-evaluator

[14] https://github.com/fusion-jena/JenTab_solution_files

[15] Values shown on a log-scale to account for large range of values.

Table 3: Metrics on each round for different modes of CTA execution. F1-Score (F1), Precision (Pr), Recall (R) and Coverage in percentage (Cov.%). AF1, APr and AR are approximated versions. Best result per task and round in **bold**.

| | | CEA | | | | CTA | | | | CPA | | | |
|---|---|---|---|---|---|---|---|---|---|---|---|---|---|
| Mode | Rounds | F1 | Pr | R | Cov.% | AF1 | APr | AR | Cov.% | F1 | Pr | R | Cov.% |
| P31 | R1 | **0.968** | 0.969 | 0.968 | 99.87 | **0.962** | 0.969 | 0.955 | 99.87 | 0.962 | 0.969 | 0.955 | **99.87** |
| | R2 | **0.975** | 0.975 | 0.975 | **99.98** | **0.965** | 0.967 | 0.962 | 99.51 | **0.984** | 0.988 | 0.979 | 99.04 |
| | R3 | 0.965 | 0.967 | 0.964 | 99.75 | **0.955** | 0.959 | 0.951 | 99.20 | **0.981** | 0.987 | 0.976 | 98.81 |
| | R4 | **0.974** | 0.974 | 0.973 | **99.39** | **0.945** | 0.941 | 0.950 | 99.31 | 0.992 | 0.994 | 0.989 | 99.51 |
| 2 Hops | R1 | 0.967 | 0.967 | 0.967 | **99.98** | 0.841 | 0.841 | 0.841 | **99.93** | **0.963** | 0.970 | 0.956 | 98.57 |
| | R2 | 0.970 | 0.970 | 0.970 | 99.97 | 0.908 | 0.909 | 0.906 | **99.73** | 0.982 | 0.987 | 0.977 | **99.06** |
| | R3 | **0.967** | 0.968 | 0.966 | 99.75 | 0.916 | 0.918 | 0.913 | **99.54** | 0.983 | 0.988 | 0.978 | **98.98** |
| | R4 | 0.973 | 0.974 | 0.973 | 93.04 | 0.930 | 0.924 | 0.937 | **99.79** | **0.993** | 0.994 | 0.992 | **99.78** |
| Multi Hops | R1 | 0.967 | 0.967 | 0.967 | **99.98** | 0.824 | 0.824 | 0.824 | 99.91 | 0.963 | 0.969 | 0.956 | 98.60 |
| | R2 | 0.966 | 0.968 | 0.965 | 99.70 | 0.927 | 0.929 | 0.925 | 99.57 | 0.982 | 0.988 | 0.976 | 98.79 |
| | R3 | 0.961 | 0.964 | 0.958 | 99.42 | 0.929 | 0.933 | 0.925 | 99.19 | 0.980 | 0.987 | 0.973 | 98.54 |
| | R4 | 0.947 | 0.949 | 0.945 | 81.83 | 0.863 | 0.892 | 0.836 | 92.23 | 0.956 | 0.994 | 0.920 | 92.63 |

Table 4: State of the art comparison. F1-Score (F1), Precision (Pr). Best result per task in **bold**. Results for other systems from [12].

| | Automatically Generated Dataset | | | | | | Tough Tables | | | |
|---|---|---|---|---|---|---|---|---|---|---|
| | CEA | | CTA | | CPA | | CEA | | CTA | |
| System | F1 | Pr | F1 | Pr | F1 | Pr | F1 | Pr | F1 | Pr |
| JenTab (P31) | 0.974 | 0.974 | 0.945 | 0.941 | 0.992 | 0.994 | 0.485 | 0.488 | 0.524 | 0.554 |
| JenTab (2 Hops) | 0.973 | 0.974 | 0.930 | 0.924 | 0.993 | 0.994 | 0.476 | 0.526 | 0.646 | 0.666 |
| JenTab (Multiple Hops) | 0.947 | 0.949 | 0.863 | 0.892 | 0.956 | 0.994 | 0.287 | 0.402 | 0.180 | 0.237 |
| MTab4Wikidata | **0.993** | **0.993** | **0.981** | **0.982** | **0.997** | **0.997** | **0.907** | **0.907** | **0.728** | **0.730** |
| bbw | 0.978 | 0.984 | 0.980 | 0.980 | 0.995 | 0.996 | 0.863 | 0.927 | 0.516 | 0.789 |
| LinkingPark | 0.985 | 0.985 | 0.953 | 0.953 | 0.985 | 0.986 | 0.810 | 0.811 | 0.686 | 0.687 |
| DAGOBAH | 0.984 | 0.985 | 0.972 | 0.972 | 0.995 | 0.995 | 0.412 | 0.749 | 0.718 | 0.747 |
| SSL | 0.833 | 0.833 | 0.946 | 0.946 | 0.924 | 0.924 | 0.198 | 0.198 | 0.624 | 0.669 |

Table 5: Execution time in days and the number of the used runners/clients for each setup mode.

| | R1 | | R2 | | R3 | | R4 | |
|---|---|---|---|---|---|---|---|---|
| Mode | Days | Runners | Days | Runners | Days | Runners | Days | Runners |
| P31 | 0.5 | 4 | 2.5 | 4 | 1.5 | 6 | 2 | 4 |
| 2 Hops | 1 | 4 | 1.2 | 4 | 2 | 4 | 1.1 | 8 |
| Multi Hops | 1 | 4 | 1.5 | 4 | 2.5 | 6 | 3.5 | 6 |

A reoccurring source of issues was the dynamic nature of Wikidata. Users enter new data, delete existing claims, or adjust the information contained. On several occasions, we investigated missing mappings of our approach only to find that the respective entity in Wikidata had changed. The challenge and ground truth were created at one point in time, so using the live system will leave some mappings unrecoverable. Moreover, we are limited by the fair-use policies of the Wikidata Endpoint service. Another limitation affects the "CEA by Column" module. Some classes like *human (Q5)* have a large number of instances. Here, queries to retrieve those instances oftentimes fail with timeouts, which limits the module to reasonably specific classes.

## 6    Conclusions and Future Work

In this paper, we presented an extensive evaluation of our toolkit for Semantic Table Annotation, "JenTab". Based purely on the publicly available endpoints of Wikidata, its modular architecture allows to exploit various strategies and easily adjust the processing pipeline. "JenTab" is publicly available[16]. We presented a detailed analyses on the effectiveness of JenTab's strategies using the benchmark dataset provided by SemTab2020. Finally, we compared JenTab to other top contenders from that challenge and demonstrate the competitiveness of our system.

We see multiple different areas for further improvement. First, certain components currently require substantial resources, either due to the number of computations necessary like the Generic Lookup or the lacking performance of the SPARQL endpoint. While we can address the latter by rewriting queries or re-designing the approach, the former offers plenty of opportunities to accelerate the system.

## Acknowledgment

The authors thank the Carl Zeiss Foundation for the financial support of the project "A Virtual Werkstatt for Digitization in the Sciences (P5)" within the scope of the program line "Breakthroughs: Exploring Intelligent Systems" for "Digitization - explore the basics, use applications". We would further like to thank K. Opasjumruskit, S. Samuel, and F. Zander for the fruitful discussions throughout the challenge. Last but not least, we thank B. König-Ries and J. Denzler for their guidance and feedback.

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
