# OpenReview forum: "JenTab: A Toolkit for Semantic Table Annotations"
_eswc-conferences.org/ESWC/2021/Workshop/KGCW — KGCW 2021_

### Official Review · ~Giuseppe_Futia1 · 2021-04-14
**A solution that achieves interesting results, but they require further discussion**

**Rating:** 7
**Confidence:** 3

**Review:**

The manuscript introduces JenTab, a modular system for annotating tabular datasets to the Wikidata Knowledge Graph. JenTab is based on an iterative approach that employs diverse contextual information to disambiguate the annotations: this procedure allows for skimming possible candidates until selecting the best match. The authors claim that their modular architecture enables a distributed processing, which allows splitting the workload across different nodes.

With regards to the positive side, the paper is well written and easy to follow. The authors' modular approach technically sounds, and the possibility of defining different settings to achieve better performances should be crucial to fit the system for different situations. The evaluation procedure is well conducted. However, the manuscript should contain a deeper discussion on the results.

Table 4 compares the results of JenTab with the SemTab2020 top systems. As claimed by the authors,  “JenTab’s results are competitive across all tasks. They are severely impacted by the Tough Tables (2T) dataset, though.” A deeper elaboration would be interesting to understand the motivations behind the JenTab results in tough tables compared to other systems. What further developments could improve the JenTab performance? Table 5 shows the execution time related to each experiment setting. What are the characteristics of the hardware used for the experiment? Another interesting discussion would be related to the impact of the modular architecture in terms of time execution compared to the SemTab2020 top systems.

In summary, the discussion on the results does not completely convince me, but I believe that the authors can defend these points if this work is presented at the workshop.

Minor:
- page 5, first bullet point: “to to”
- page 6, CTA support bullet point: “that to not apply”

---

### Official Review · ~Ernesto_Jiménez-Ruiz2 · 2021-04-14
**A good system**

**Rating:** 7
**Confidence:** 5

**Review:**

The paper presents a modular and open source system (JenTab) for the semantic annotation of tabular data. JenTab represents a very relevant engineering effort to gather good practices to annotate tabular data and a way that seems easy to customize. What I'm missing is some additional information about the novel research questions that JenTab is trying to answer. From the research point of view, what is making JenTab different? My suggestion is that JenTab should probably  go in depth in some of its modules and include a novel technique.

The paper also presents a good overview of typical challenges. The related work is perhaps too focused on current SemTab participants.

The evaluation is interesting and provides insights about the different modules in JenTab. The results are competitive although lower than expected in the 2T dataset.
Why, according to the authors, the results in 2T are much lower?

As the P31 strategy is mentioned, I'm wondering if JenTab can work with target KGs other than Wikidata.

Other comments:
- "The  tables  in  the  SemTab2020  datasets  [8, 10]  are artificially created from Wikidata"-->Not all, the tough tables dataset used in Round 4 has been manually curated (see [a]).

- "Colnet : Results  have  shown  that  the  lookup  service  outperforms  the CNN prediction for a larger knowledge gap." --> it is actually the opposite. Look-up works very well (even better than CNN) for cases without ambiguity or knowledge gap, however the benefits of the CNN appear for more complex tables.

- ptype may be a good alternative to identify the datatype of a column [b].

Minor:
- Page 3:
is 8 is --> footnote is misplaced
SemTab2020. --> SemTab2020,
- Page 11:
add meaning of P31 and P279
- Figure 7 and 8.
The y-labels seem to be wrong (i.e., no log-scale)


[a] Vincenzo Cutrona, Federico Bianchi, Ernesto Jiménez-Ruiz and Matteo Palmonari.Tough Tables: Carefully Evaluating Entity Linking for Tabular Data. International Semantic Web Conference (ISWC). 2020. https://openaccess.city.ac.uk/id/eprint/24776/1/Tough_Tables_Carefully_Evaluating_Entity_Linking_for_Tabular_Data.pdf
[b] ptype: https://tahaceritli.github.io/software/

---

### Official Review · ~Boris_Villazon-Terrazas1 · 2021-04-15
**Good paper**

**Rating:** 7
**Confidence:** 3

**Review:**

This paper introduces an approach to perform semantic table annotations.
The authors presented a good related work about the available tools that perform the semantic table annotations. Moreover, they present their approach along with an initial evaluation.
I would like to see their presentation during the workshop.

Comments
---------------
- In spite the authors pointed out some drawbacks of the related work, I would like to see what are the major/minor differences with the proposed approach; in terms of method and techniques.
- Within the their approach authors rely on Levenshtein distance to perform string similarity. I would like to see the reason behind that. did the authors try other distances, for example Jaro/JaroWinkler, which are good for short strings?
Moreover, the authors probably can try other libraries such as facebook duckling ```[1]

I am quite busy, so I was not able to dig in more on the code, and run your project; but I will at some point.

`[1] https://github.com/facebook/duckling

---

### Official Review · ~Francesco_Osborne1 · 2021-04-15
**A fair contribution**

**Rating:** 7
**Confidence:** 4

**Review:**

The paper presents JenTab, a new system for the semantic annotation of tables.

The paper is fairly written and clear. JenTab appears to be a useful tool that may be reused by the community. However, the description of the approach is a bit superficial. In particular, section 3.2 should give more details about how mappings are created. The state of the art could be also improved, but it is fine for a workshop paper.

The evaluation is well done and comprehensive. JenTab is competitive, but does not to obtain state of the art results and performs particularly bad on the 'tough tables' dataset. The authors should discuss more extensively the results and in particular the problematic performance on 'tough table'. It would be also useful to add some more details about scalability.

The main limitation of the paper is that the research contribution is not clear. JenTab seems to be similar to other systems in the same domain and I do not understand what the added value is. However, I think that this work has potential, and it is a good contribution for the workshop.

---

### Meta-Review · Program_Chairs · 2021-04-20

**Recommendation:** Accept
**Confidence:** 5

**Metareview:**

Dear authors,

Your work introduces a new system that has potential for our community. But there are some important aspects that would improve the quality of the paper:

- What is the added value of your system compared to existing systems?
- Add more explanations and details to the results.

---

### Decision · Program_Chairs · 2021-04-23

Accept